# Inversion Analysis of Impervious Curtain Permeability Coefficient Using Calcium Leaching Model, Extreme Learning Machine, and Optimization Algorithms

**Yongkang Shu [1], Zhenzhong Shen [1,2], Liqun Xu [1,*], Kailai Zhang [1] and Chao Yang [1]**

1   College of Water Conservancy and Hydropower Engineering, Hohai University, Nanjing 210098, China; shuyongkang@hhu.edu.cn (Y.S.); zhzhshen@hhu.edu.cn (Z.S.); zhangkailai@hhu.edu.cn (K.Z.); yangchao96@hhu.edu.cn (C.Y.)
2   State Key Laboratory of Hydrology-Water Resources and Hydraulic Engineering, Hohai University, Nanjing 210098, China
*   Correspondence: xlq@hhu.edu.cn

**Abstract:** The calcium leaching effect leads to a decrease in the impermeability of the impervious curtain. The inverse analysis strategy was introduced in this study because the calcium leaching parameters of the curtain are not easy to determine. An orthogonal design and the finite element method were used in the strategy. The time-series data of hydraulic head and leakage volume were applied to construct the objective function. The extreme learning machine (ELM) was proposed to build the reflection sets. Genetic algorithm (GA), simulated annealing (SA), sparrow search algorithm (SSA), and particle swarm optimization (PSO) were employed to accelerate the iterative search for the target parameters. The target parameters of the calcium leaching model were used for finite element verification by comparing the monitored and simulated values. The simulated values of hydraulic head and leakage by PSO show good agreement with measurements. The evolution of the curtain permeability coefficient in 100 years was analyzed. The results demonstrate the strategy's feasibility in determining the curtain's calcium leaching parameters and permeability coefficients.

**Keywords:** inverse analysis; calcium leaching; impervious curtain; permeability coefficient

## 1. Introduction

The curtain is an impermeable structure used for dam foundation containment in hydraulic engineering. Calcium hydroxide (CH) and calcium silicate hydrate (C-S-H) are the main components of the solid-phase calcium in the curtain. CH partly determines the decay process of the curtain's impermeability. At the same time, C-S-H affects the physical and mechanical properties of the curtain. The precipitation of the solid-phase calcium in cement-based materials occurs in weakly alkaline or basic solutions. During operation, the cyclic action of ambient water causes the solid-phase calcium in the curtain to decompose and precipitate out, increasing the permeability of the curtain and affecting the operation of the hydraulic project [1]. For example, after nearly twenty years of operation, the Daheiding dam had a significant calcium precipitation problem, deteriorating year by year. The impervious curtain was found to be seriously damaged by sampling the curtain bore-hole and underwater television observation [2].

The permeability coefficient of cement-based materials is generally defined as a function of porosity, influenced by pore structure, degree of dissolution, and physical damage [3]. Saito [4] proposed an exponential equation between permeability and porosity for simulating the permeability coefficient of cement-based materials during leaching. This equation works effectively in accelerated mortar electrochemical tests and has been adopted by Gawin [5,6]. Based on this, Kozeny and Carman further considered the evolution of microstructural parameters by introducing pore shape, specific surface area, and degree

of distortion. This evolution provides a more accurate characterization of the hydraulic properties and has been extensively used in species of cement-based materials [7–10]. Significantly, the permeability coefficient of cement-based materials is similarly influenced by the process and rate of solid-phase calcium decomposition. For diffusion-driven leaching, Gerard et al. [11], Phung et al. [12], and Wan et al. [13,14] have proposed different solid–liquid phase equilibrium equations based on thermodynamic equilibrium relations. Regarding advection-diffusion-driven leaching, Lambert [15] adopted the discrete element method to simulate the dissolution process at the rock–mortar contact, which provided a new strategy for solving the calcium leaching problem. Zhang et al. [16] considered the calcium leaching effect and presented the calcium hydroxide content and initial infiltration flow rate as the durability control index of the impervious curtain.

A great deal of research has been conducted on the calcium leaching effect. However, one primary problem with calcium leaching is that those calculation parameters are not efficiently and accurately determined. This problem is accentuated by the little research on calcium leaching parameters during the construction period.

An alternative approach to the problem is inversion analysis. The inversion method has been proven to be an efficient way for obtaining rational seepage parameters [17–24]. Nevertheless, little research has been carried out specifically for the permeability coefficient of the impervious curtain. Furthermore, the existing studies simply considered the role of the seepage field without applying the coupling model of the chemical and seepage fields [25,26], which is unable to determine the parameters accurately for the calcium leaching problem.

The essence of inversion analysis is to determine the calculation parameters based on observations. The inversion analysis exists relative to the forward computation. An artificial neural network is required to construct the data mapping in the forward research, while an optimization algorithm is selected to minimize the objective function in the back process. The optimal parameters are found by iterative computation while minimizing the defined objective function. The results are then substituted into the finite element model, and the error between the calculated results and measurements is evaluated accordingly.

Due to the relatively time-saving and inexpensive availability of head data in hydraulic engineering, the root means square of the head error is simply set to the objective function in most inversion models [27–30]. One limitation of this strategy is that it could lead to non-uniqueness problems [31]. The effectiveness of the inversion model can be improved by combining several types of observations (e.g., hydraulic head and leakage).

In this paper, the inverse analysis strategy is used for obtaining calcium leaching parameters by considering the coupling of the seepage and chemical fields. The objective function is constructed by time-series measurements. Finite element analysis, orthogonal design, and extreme learning machine (ELM) are taken to determine the objective parameters. Genetic algorithm (GA), simulated annealing (SA), sparrow search algorithm (SSA), and particle swarm optimization (PSO) are used to speed up the search process, respectively. The strategy is applied to the inverse model of the Shimantan concrete gravity dam located in Henan Province, China. Combined with the observed data and analytical methods, the optimal calcium leaching parameters are determined, and the curtain permeability coefficient evolution in a century is presented.

## 2. Calcium Leaching Model

### 2.1. Controlling Equations

Impervious curtains consist of pouring materials such as cement mortar and clay slurry. The application of curtain grouting is an important seepage control measure in water engineering. Under prolonged exposure to ambient water, the solid-phase calcium in the curtain (e.g., CH and C-S-H) decomposes and precipitates because of the leaching effect. The decomposition of the solid-phase calcium leads to the increasing porosity and permeability coefficient of the materials. Assuming that the flow of the pore solution follows

Darcy's law, the equations controlling permeation dissolution in curtains are presented in Equation (1).

$$
\begin{cases}
u = -\frac{k}{\rho g}\nabla P \\
\frac{\partial(\varphi P)}{\partial t} + \nabla(\rho u) = Q_m \\
\frac{\partial c}{\partial t} + \nabla(-D\nabla c) + u\nabla c = R_C
\end{cases}
\tag{1}
$$

where $u$ denotes the osmotic flow rate; $k$ is the permeability coefficient; $\rho$ reprents the water density; $g$ means the gravitational acceleration; $\varphi$ is the porosity; $P$ stands for the water pressure; $t$ represents the time; $Q_m$ denotes the mass source items; $c$ is the concentration of $Ca^{2+}$ in the pore solution; $D$ represents the diffusion coefficient; and $R_C$ means the rate of solid-phase calcium decomposition.

### 2.2. The Solid-Phase Calcium Decomposition Model

The solid–liquid equilibrium equation cannot be applied directly to simulate the decomposition of solid-phase calcium in advection-diffusion-driven leaching. In this case, Ulm et al. [32] provided a chemical pore plasticity theory that quantifies the rate of solid-phase calcium decomposition by the distance from equilibrium. The disintegration of the solid-phase calcium can be expressed as Equation (2) after Gawain's neglect of the elastic deformation and plastic hardening–softening term.

$$
\begin{cases}
\frac{\partial s_{Ca}}{\partial t} = \frac{1}{\eta}A_s \\
\eta = RT\tau_{leach} \\
A_s = RT\ln\left(\frac{c_{Ca}}{c_{Ca}^{eq}}\right) - \int_{S_{Ca}^{eq}}^{S_{Ca}}\kappa(\bar{s})\mathrm{d}\bar{s} \\
\kappa(\bar{s}) = \frac{RT}{c_{Ca}}\left(\frac{\mathrm{d}s_{Ca}}{\mathrm{d}c_{Ca}}\right)^{-1}
\end{cases}
\tag{2}
$$

where $s_{Ca}$ represents the actual calcium concentration in the solid skeleton; $t$ denotes the time; $\eta$ is the micro-diffusion of $Ca^{2+}$ in different compounds, depending on the calcium content; $R$ means the gas constant; $T$ is the temperature; $\tau_{leach}$ denotes the characteristic time of calcium leaching; $A_s$ represents the chemical affinity that controls the force of chemical reactions; $c_{Ca}$ stands for the present calcium concentration in the pore fluid; $\kappa(\bar{s})$ is the equilibrium constant; and $c_{Ca}^{eq}$ and $s_{Ca}^{eq}$ mean the calcium concentration in the pore fluid and solid skeleton at the equilibrium point, respectively.

### 2.3. Pore Parameter Evolution Equation
### 2.3.1. Kozeny–Carman Equation

The Kozeny–Carman (KC) equations are the best known semi-empirical formulation in the field of percolation and they have been adopted extensively in the simulation of permeability coefficient. Considering the effects of porosity, pore internal surface area, and distortion, the permeability coefficient of cement-based materials can be described by the KC equation in Equation (3).

$$
K = \frac{\varphi^3}{c(1-\varphi)^2 S^2}
\tag{3}
$$

where $K$ means the permeability coefficient of cement-based material; $\varphi$ is the porosity; $c$ denotes the Kozeny–Carman constant; and $S$ represents the specific surface area of the solid phase.

The parameter $c$ in the KC equations can be replaced by $\tau^2 F_s$ for simplifying the effect of pore shape and degree of distortion. The expression is given in Equation (4).

$$
\begin{cases}
\Omega = \frac{1}{\tau^2 F_s} \\
\Omega_0 = \frac{1}{n}\Omega_l \\
\Omega = \Omega_0 - (\Omega_0 - \Omega_l)d_l^2
\end{cases}
\tag{4}
$$

where $\Omega_0$ and $\Omega_l$ denote the set total term for undissolved and dissolved material, respectively; $\tau$ is the tortuosity; $F_s$ stands for the shape factor; $n$ means the lumped term increased times; $d_l$ represents the leaching degree. The leaching degree is given as Equation (5).

$$d_l = \begin{cases} 1 & C_{CH} = 0 \\ \frac{C_{CH}}{C_{CH}^0} & C_{CH} > 0 \end{cases} \tag{5}$$

where $C_{CH}^0$ and $C_{CH}$ denotes the initial and current concentration of CH, respectively.

### 2.3.2. Porosity Variation Equation

In advection-diffusion-driven leaching, the dissolution of solid-phase calcium leads to increased porosity. Combining the coupled permeation–dissolution model for porous media materials with the simplified method of calculating porosity proposed by Kuhl [33,34], the evolution equation for porosity is presented in Equation (6).

$$\varphi = \varphi_0 + \frac{u}{\rho} \int \frac{1}{\eta} A_s dt \tag{6}$$

where $\varphi$ and $\varphi_0$ represent the current and initial porosity of the material, respectively; $\frac{u}{\rho}$ means the average molar volume of the solid phase, which is taken as 0.056 mol/m$^3$ in this paper; $A_s$ is the chemical potential; and $\eta$ denotes the coefficient affecting the micro-diffusion of the Ca$^{2+}$ in the pores. The values of different $\eta$ are indicated in the finite element analysis part.

### 2.3.3. Diffusivity Evolution Equation

The diffusivity of cement-based material rises continuously with increasing porosity in advection-diffusion-driven leaching. Van Eijk and Brouwers [35] proposed a modified equation for the relationship between porosity and diffusivity. The connection is given in Equation (7).

$$\begin{aligned} \frac{D_e}{D_0} &= 0.0025 - 0.07\varphi_c(x,0)^2 - 1.8H(\varphi_c(x,0) - 0.18)(\varphi_c(x,0) - 0.18)^2 \\ &\quad + 0.14\varphi_c(x,t)^2 + 3.6H(\varphi_c(x,t) - 0.16)(\varphi_c(x,t) - 0.16)^2 = D(\varphi) \end{aligned} \tag{7}$$

where $D_0$ represents the aquatic diffusivity of Ca$^{2+}$ which is taken as $4.5 \times 10^{-10}$ m/s; $\varphi_c(x,0)$ means the initial capillary porosity; $H()$ is the Heaviside function; and $\varphi_c(x,t)$ denotes the capillary porosity.

## 3. The Objective Function

Recent theoretical developments have revealed that multiple types of observations can identify the objective function of inversion analysis. In this study, the hydraulic head and leakage measurements are applied to construct the objective function. It is assumed that the permeability coefficient of the rock stratum is isotropic. $P_i^m = \left[ P_{i1}^m, P_{i2}^m, P_{i3}^m, \ldots P_{it}^m \right] (i = 1, 2, 3, \ldots, M)$ denotes the time series data for pressure tube monitoring, while $Q_j^m = \left[ Q_{j1}^m, Q_{j2}^m, Q_{j3}^m, \ldots Q_{jt}^m \right] (j = 1, 2, 3, \ldots, N)$ denotes the time series data for weir monitoring. A scalar K was selected to stand for the permeability coefficient of the impervious curtain.

The permeability coefficient of the curtain increases with calcium leaching time. The KC equations were used to simulate the evolution of the permeability coefficient under the influence of calcium leaching. The porosity was applied as an intermediate parameter linking the calcium leaching effect to the permeability coefficient. In the calcium leaching model, the lumped term for sound materials $\Omega_0$, the lumped term increased times $n$, and the rock hydraulic conductivity $k_r$ have significant implications on the permeability coefficient of the curtain. In this paper, the inverse model was developed to obtain

the best estimates of these parameters to minimize the value of the objective function F. The mathematical model of the back analysis is established as described in Equation (8):

$$\min F = \left( \sum_{i=1}^{M} \frac{||P_i(K(\Omega_0,n,k_r,t))-P_i^m||_2^2}{||P_i^m||_2^2} \right)^{\frac{1}{2}} + w \left( \sum_{j=1}^{N} \frac{||Q_j(K(\Omega_0,n,k_r,t))-Q_j^m||_2^2}{||Q_j^m||_2^2} \right)^{\frac{1}{2}}$$
$$\text{s.t.} \ \ \Omega_{0min} \le \Omega_0 \le \Omega_{0max}$$
$$n_{min} \le n \le n_{max}$$
$$k_{rmin} \le k_r \le k_{rmax}$$

(8)

Here $P_i(K(\Omega_0,n,k_r,t))$ and $Q_j(K(\Omega_0,n,k_r,t))$ are numerically calculated hydraulic head and leakage volume, respectively, and $w$ is the weight for ensuring a relative balance between the head and flow observation errors. Zhou et al. [17] analyzed the relative error of hydraulic head and leakage under different weight values. The results show that the errors between hydraulic head and discharge can be well balanced under the condition of $w = 0.02$. The conclusion is adopted in this study. $\Omega_{0max}$ and $\Omega_{0min}$ respectively represent the upper and lower bounds of the lumped term. $n_{max}$ and $n_{min}$ denote the upper and lower bounds of the lumped term increased times. Similarly, $k_{rmax}$ and $k_{rmin}$ signify the upper and lower bounds of rock hydraulic conductivity, respectively. The upper and lower bounds of the parameters can be roughly determined by engineering experience and laboratory experiments. The ranges of parameters are indicated in the results section, for verifying the rationality of the results.

Within the range of the monitoring data, the hydraulic head and leakage measurements at the initial moment were selected for inversion to obtain the target parameters. The combination of the objective parameters was also applied to predict the trend of curtain permeability coefficient in a century.

## 4. Prediction Model of Permeability Coefficient

### 4.1. Orthogonal Design Method

OD is an experimental method for studying multiple factors and levels [36]. The selection of representative combinations within comprehensive test portfolios can be efficient and economical. Suppose that there is no interaction between parameters. The typical combinations of parameters are selected for the FEM positive computation. For the parameters $\Omega_0$, $n$, and $k_r$ in this paper, five uniformly distributed values were chosen for each of the range of values taken. Under a full-scale test, 125 ($5^3$) trials are required, and no replications of each combination are considered. By contrast, only 25 ($L_{25}(5^3)$) trials are required by choosing the OD method. OD dramatically reduces the number of trials and constructs representative samples. The orthogonal combinations used in this paper are given in Table 1.

**Table 1.** Sample parameters based on orthogonal design $L_{25}(5^3)$.

| Test Number | Permeation Parameters | | | Adaptability Value |
|:---:|:---:|:---:|:---:|:---:|
| | $\Omega_0$ | $n$ | $k_r$ | |
| 1 | 5000 | 500 | $1 \times 10^{-7}$ | 0.004774 |
| 2 | 5000 | 800 | $4.6 \times 10^{-7}$ | 0.244917 |
| 3 | 5000 | 1100 | $8.2 \times 10^{-7}$ | 0.537412 |
| 4 | 5000 | 1400 | $2.8 \times 10^{-7}$ | 1.086679 |
| 5 | 5000 | 1700 | $6.4 \times 10^{-7}$ | 1.409979 |
| 6 | 8000 | 500 | $8.2 \times 10^{-7}$ | 0.01401 |
| 7 | 8000 | 800 | $2.8 \times 10^{-7}$ | 0.371828 |
| 8 | 8000 | 1100 | $6.4 \times 10^{-7}$ | 1.055184 |
| 9 | 8000 | 1400 | $1 \times 10^{-7}$ | 1.258973 |
| 10 | 8000 | 1700 | $4.6 \times 10^{-7}$ | 1.458692 |

**Table 1.** *Cont.*

| Test Number | Permeation Parameters | | | Adaptability Value |
|---|---|---|---|---|
| | $\Omega_0$ | *n* | $k_r$ | |
| 11 | 11,000 | 500 | $6.4 \times 10^{-7}$ | 0.019933 |
| 12 | 11,000 | 800 | $1 \times 10^{-7}$ | 0.640415 |
| 13 | 11,000 | 1100 | $4.6 \times 10^{-7}$ | 1.239977 |
| 14 | 11,000 | 1400 | $8.2 \times 10^{-7}$ | 1.636077 |
| 15 | 11,000 | 1700 | $2.8 \times 10^{-7}$ | 2.748317 |
| 16 | 14,000 | 500 | $4.6 \times 10^{-7}$ | 0.035461 |
| 17 | 14,000 | 800 | $8.2 \times 10^{-7}$ | 0.696786 |
| 18 | 14,000 | 1100 | $2.8 \times 10^{-7}$ | 1.298395 |
| 19 | 14,000 | 1400 | $6.4 \times 10^{-7}$ | 2.484507 |
| 20 | 14,000 | 1700 | $1 \times 10^{-7}$ | 2.892282 |
| 21 | 17,000 | 500 | $2.8 \times 10^{-7}$ | 0.038281 |
| 22 | 17,000 | 800 | $6.4 \times 10^{-7}$ | 0.702525 |
| 23 | 17,000 | 1100 | $1 \times 10^{-7}$ | 1.879253 |
| 24 | 17,000 | 1400 | $4.6 \times 10^{-7}$ | 2.515877 |
| 25 | 17,000 | 1700 | $8.2 \times 10^{-7}$ | 4.068743 |

*4.2. Extreme Learning Machine*

ELM is a general estimation of multivariate nonlinear mapping with good generalization performance and learning ability. As a prediction tool, ELM has been extensively used in hydrological and geological problems [37–39]. The weights between the input and hidden layers are randomly set and kept constant in the ELM strategy, eliminating the back-propagation operation process. At the same time, the system of solving equations is applied to directly determine the connection weights between the implicit layer and the output layer, which significantly improves the generalization ability and learning speed of the model. The input part of the ELM model training sets consists of calcium leaching parameters determined by OD. The output includes the combined head and leakage errors calculated by the FEM simulation at the measurement points.

Figure 1 shows a neural network with a single hidden layer, where $I_i (i = 1, 2, 3)$ represents the input layer. $S_i (i = 1, 2, 3)$ means the single-hidden layer. O is the output layer. $\Omega_0$, *n*, and $k_r$ are the calcium leaching parameters. $a_{ij} = [a_{i1}, a_{i2}, a_{i3}] (i = 1, 2, 3)$ denotes the input weight of the $i^{th}$ implied layer unit and $v_i$ means the weight of the $i^{th}$ implied layer output unit. The output layer contains only one unit because of the integration between hydraulic head and leakage. One point to note is that there may be situations where the matrix cannot be inverted since the mapping function is initialized randomly. To solve this problem, the mapping function is chosen to be a Sigmoid function to ensure that the output matrix of the hidden layer achieves full row rank or full column rank.

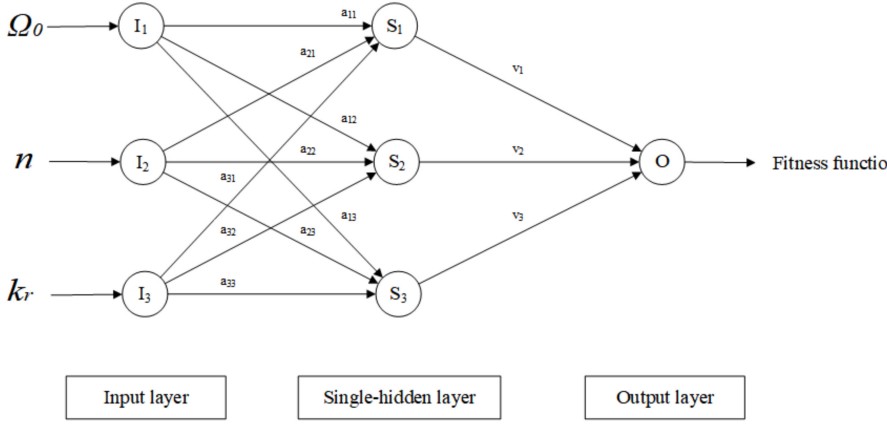

**Figure 1.** A single implicit layer ELM neural network.

*4.3. Predictive Modeling Procedure*

The steps of the inversion model constructed by the ELM and optimization algorithms are given in the following steps, and the flow chart is shown in Figure 2:

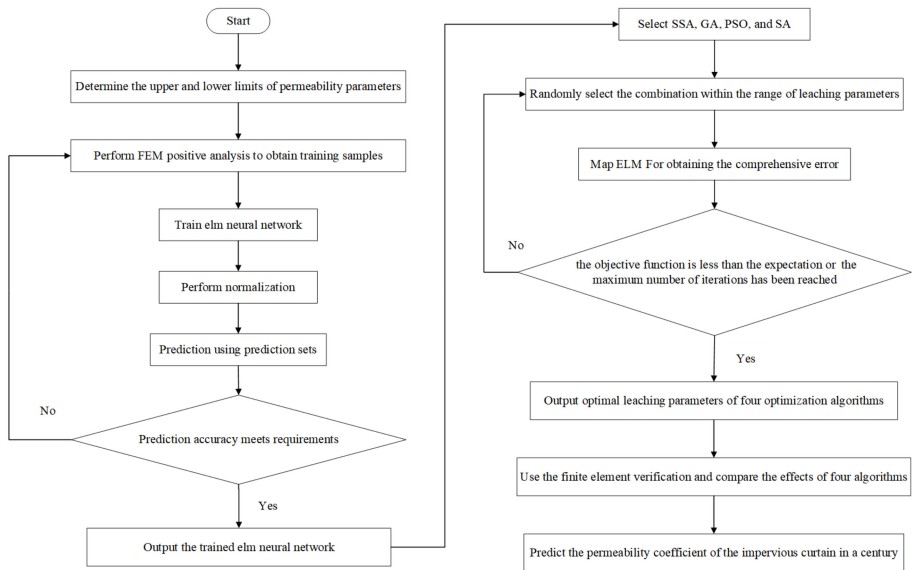

**Figure 2.** Flow chart of the prediction model based on ELM and optimization algorithms.

Step 1: Determine the upper and lower limits of each leaching parameter. Select reasonable combinations of parameters by the orthogonal design. Carry out a positive computation by the finite element method, obtain the hydraulic head and leakage values of the measurement points, and calculate the corresponding error values.

Step 2: The combinations of leaching parameters are used as inputs to the ELM, and the error values of measurement points are selected as outputs. Establish a nonlinear mapping between the parameter combinations and the error values.

Step 3: Four optimization algorithms are chosen to obtain the objective combination that minimizes the objective function. The corresponding error value is determined by mapping the ELM.

Step 4: Determine whether the maximum number of iterations has been reached or whether the target function meets the accuracy requirements. If not, return to Step 3; otherwise, output global optimal solution and fitness value.

Step 5: The objective combination of leaching parameters is substituted into the finite element model to calculate the hydraulic head and leakage volume at the measurement points. Compare the simulation results with monitoring data and evaluate the accuracy of the objective combination. Predict the permeability coefficient of the impervious curtain in a century.

## 5. Application Case

*5.1. Project Overview*

The two-dimensional finite element model of a gravity dam was established for verifying the effectiveness of the inversion strategy. Based on the geological investigation and monitoring analysis, the inverse calculation of the calcium leaching parameters was carried out with ELM and four optimization algorithms.

Located on the Rolling River in the Henan province of China, the Shimantan dam is a large-scale water conservancy project with integrated use of industrial water supply, flood control, flood removal, and irrigation. The location of the Shimantan gravity dam is presented in Figure 3.

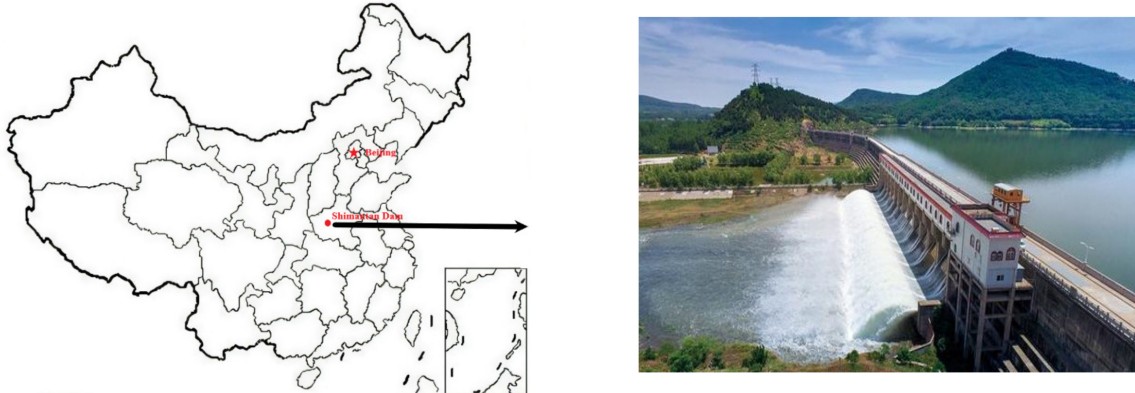

**Figure 3.** The location of the Shimantan gravity dam.

The dam mainly consists of secondary with crushed concrete, tertiary with crushed concrete, normal concrete, and impermeable curtain. The maximum height is 40.5 m, and the maximum width is 31.74 m. The normal storage level is 107 m. The grouted drainage gallery is located in the dam near the upstream side with a bottom elevation of 76.00 m and a city gate cavern type cross-section.

### 5.2. Finite Element Analysis

#### 5.2.1. Finite Element Model

The seepage process at the base of the gravity dam was simulated by the multi-physics field simulation software COMSOL Multiphysics. Four-node convention/diffusion quadrilateral elements are applied in this study. Figure 4 shows the element meshes of the model. The model contains 4157 quadrilateral meshes. The curtain grouting thickness is 2 m, reaching a slightly weathered rock of 3 m. The maximum cell width is 2.5 m, and the minimum cell width is 0.1 m. A right-angle coordinate system was constructed, with the x-positive direction pointing downstream and the z-positive direction pointing vertically upwards. Two times the maximum height of the dam was extended at both the upstream and downstream in the x-direction. The depth of the foundation was taken as twice the dam height. The 2D finite element model is shown in Figure 4.

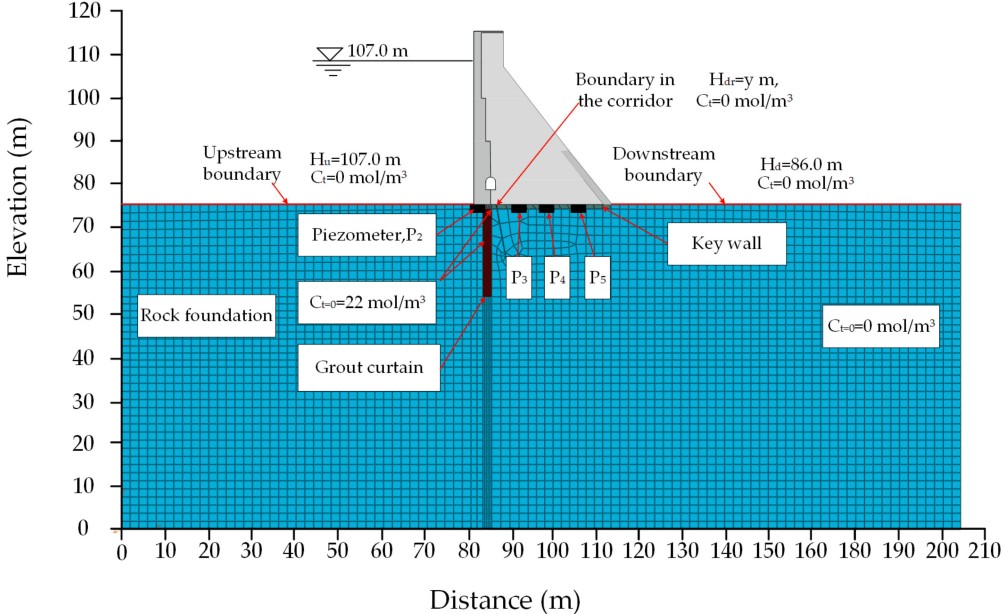

**Figure 4.** 2D finite element mesh of Shimantan dam.

### 5.2.2. Boundary and Initial Conditions

The boundary and initial conditions of the finite element model are indicated in Figure 4. The upstream and downstream water levels are 107 m and 86 m, respectively. The pore fluid within the curtain is assumed to be saturated initially. The initial concentration of calcium ions in the impervious curtain and rock foundation are 22 mol/m$^3$ and 0 mol/m$^3$, respectively.

### 5.2.3. Calculation Parameters

The initial composition of the curtain rub material could not be accurately determined as the Shimantan dam has been in operation for over twenty years. Therefore, the parameters of Sample 3 were used for some of the curtain parameters in this study. Suppose that the parameters of non-equilibrium solid–liquid dissolution followed Gawin's model. The parameters of the bedrock and curtain are shown in Table 2. The calculation parameters of the model are indicated in Table 3.

**Table 2.** Calculation parameters of rock and grout curtain.

| Material | Parameter | Notation | Value |
|---|---|---|---|
| Rock | Bulk density | $\gamma_r$ | 25.40 kN/m$^3$ |
| Rock | Initial porosity | $\varphi_f$ | 0.10 |
| Rock | Initial diffusivity | $D_{r0}$ | $1.47 \times 10^{-11}$ m$^2$/s |
| Concrete | Bulk density | $\gamma_c$ | 23.51 kN/m$^3$ |
| Concrete | CH content | $C_{c\_CH}$ | 3027 mol/m$^3$ |
| Concrete | CSH content | $C_{c\_CSH}$ | 6054 mol/m$^3$ |
| Concrete | Initial porosity | $\varphi_{c0}$ | 0.10 |
| Concrete | Initial diffusivity | $D_{c0}$ | $7.10 \times 10^{-12}$ m$^2$/s |
| Impervious curtain | CH content | $C_{CH}$ | 3027 mol/m$^3$ |
| Impervious curtain | CSH content | $C_{CSH}$ | 6054 mol/m$^3$ |
| Impervious curtain | Initial porosity | $\varphi_0$ | 0.15 |
| Impervious curtain | Initial diffusivity | $D_0$ | $9.87 \times 10^{-12}$ m$^2$/s |
| Impervious curtain | Intact/leached bulk density | $\rho_0/\rho_L$ | 30.6/145.8 |

**Table 3.** Parameters for the solid-phase calcium disequilibrium decomposition model.

| Skeleton Compound | Ca$^{2+}$ (mol/m$^3$) | d$s_{Ca}$/d$c_{Ca}$ | Diffusivity (m$^2$/s) | $\tau_{leach}$ (s) | $\frac{1}{\eta}$ (mol/(J·s)) |
|---|---|---|---|---|---|
| CH | 19~22 | 2142 | $1.47 \times 10^{-9}$ | $1.17 \times 10^4$ | $3.45 \times 10^{-8}$ |
| C-S-H | 2~19 | 203 | $1.62 \times 10^{-9}$ | $5.88 \times 10^2$ | $7.00 \times 10^{-9}$ |
| C-S-H | 0~2 | 1910 | $1.83 \times 10^{-9}$ | $6.52 \times 10^3$ | $6.20 \times 10^{-8}$ |

### 5.3. Results of the Simulation

#### 5.3.1. The Simulated Parameters

The iterative search process for the optimal calcium leaching parameters was carried out through GA, SA, SSA, and PSO, respectively. In the GA strategy, the population size is 40 and the termination evolution algebra is 200. The crossover and mutation probability are 0.7 and 0.01, respectively. In the SA strategy, the maximum number of iterations is 100, the temperature attenuation coefficient is 0.95, the initial temperature is 100, and the minimum temperature is $1 \times 10^{-6}$. In the SSA strategy, the number of sparrows is 50, the proportion of discoverers is 0.7, the proportion of followers is 0.1, the proportion of vigilants is 0.2, the maximum number of iterations is 200, and the safety threshold is 0.6. In the PSO strategy, the updated speed of the particle is the sum of its own speed inertia, self-cognition, and social cognition in the previous step. The initial population number is 20, the maximum number of iterations is 200, and the inertia weight is 0.8. The first and second learning factors are 1.5 and 1.5, respectively.

Table 4 lists the calculated calcium leaching parameters of the impervious curtain at the initial moment. The calculated values of $\Omega_0$, $n$, and $k_r$ are within the corresponding parameter ranges.

**Table 4.** Results of the simulated parameters for four algorithms.

| Optimization Algorithm / Inversion Parameter | $\Omega_0$ | $n$ | $k_r$ $(m/s)$ |
|---|---|---|---|
| GA | 9334 | 1029 | $5.36 \times 10^{-7}$ |
| SA | 13,583 | 1036 | $1.53 \times 10^{-7}$ |
| SSA | 12,756 | 1225 | $1.48 \times 10^{-7}$ |
| PSO | 12,253 | 1594 | $8.83 \times 10^{-8}$ |
| Range of parameter | 5000~17,000 | 500~1700 | $1.0 \times 10^{-8}$~$1.0 \times 10^{-6}$ |

### 5.3.2. Simulation Results of Hydraulic Head and Leakage

For verifying the validity of the simulations and the reasonableness of the calculated results, the leaching parameters determined by the four algorithms were substituted into the finite element model. The variations in the hydraulic head at measurement point $P_4^9$ and dam base leakage between 2013 and 2010 were calculated. The monitoring data documented the actual hydraulic head and dam foundation leakage changes in these eight years. The position of the piezometer $P_4^9$ is presented in Figure 1.

The leakage quantity of the dam foundation is roughly estimated by Equation (9).

$$q_b = \iint vL_0 dx dz \tag{9}$$

where $q_b$ is the leakage quantity of the dam foundation; $v$ denotes the flow velocity perpendicular to the overflow surface; and $L_0$ represents the total length of the dam, taken as 645 m.

Figure 5a shows the comparison between the calculated and monitored heads at the piezometer $P_4^9$. Linear fitting was carried out according to the observed values of the hydraulic head. The hydraulic heads determined by PSO show the best consistency with the monitoring data, while the results simulated by GA give the worst performance. The poor performance of GA may be due to the easy convergence of the algorithm to the local optimal solutions. The results of PSO embody the high accuracy in this analysis and demonstrate the validity of the simulation.

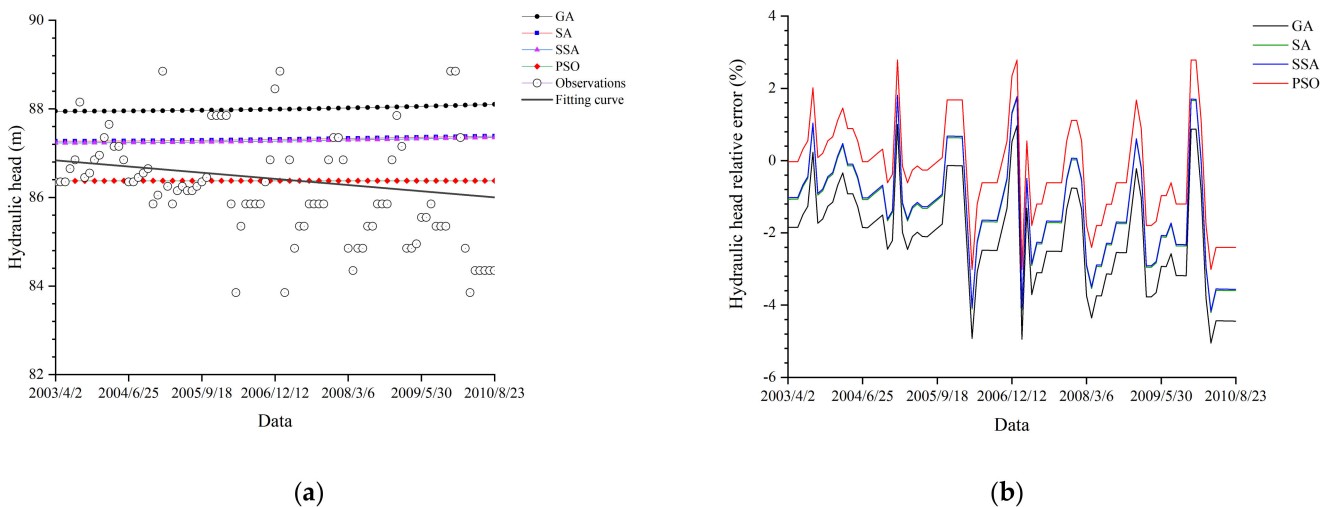

**Figure 5.** Comparison of measured and simulated values at piezometer $P_4^9$: (**a**) hydraulic head; (**b**) hydraulic head relative error.

The relative error of the hydraulic head is calculated by Equation (10).

$$\delta_P = \frac{P - \overline{P}}{\overline{P}} \times 100\% \tag{10}$$

where $\delta_P$ is the relative error of hydraulic head; $P$ represents the monthly average of predicted hydraulic head; and $\overline{P}$ means the monthly average of measured hydraulic head.

The annual average head at the piezometer $P_4^9$ was applied to calculate the relative error values. Figure 5b shows the relative error line graph of the four algorithms at the piezometer $P_4^9$ from 2003 to 2010. The maximum value of relative error in the hydraulic head is −5% in 2007. Among the four algorithms, PSO reflects the slightest relative error fluctuation. The hydraulic head relative error corresponding to PSO is between −3% and 3%.

Figure 6a presents the comparison between the calculated and measured dam foundation leakage by the four algorithms. Similarly, a fitting curve was made according to the observed leakage of the dam foundation. The simulated results by PSO agree well with the measurements, illustrating the accuracy of the simulation. The results determined by SA and SSA show medium accuracy in the analysis. The simulated results by GA deviate furthest from the monitoring data.

The relative error of leakage quantity is calculated by Equation (11).

$$\delta_Q = \frac{Q - \overline{Q}}{\overline{Q}} \times 100\% \tag{11}$$

where $\delta_Q$ is the relative error of hydraulic head; $Q$ means the annual average of predicted leakage; and $\overline{Q}$ denotes the annual average of measured leakage.

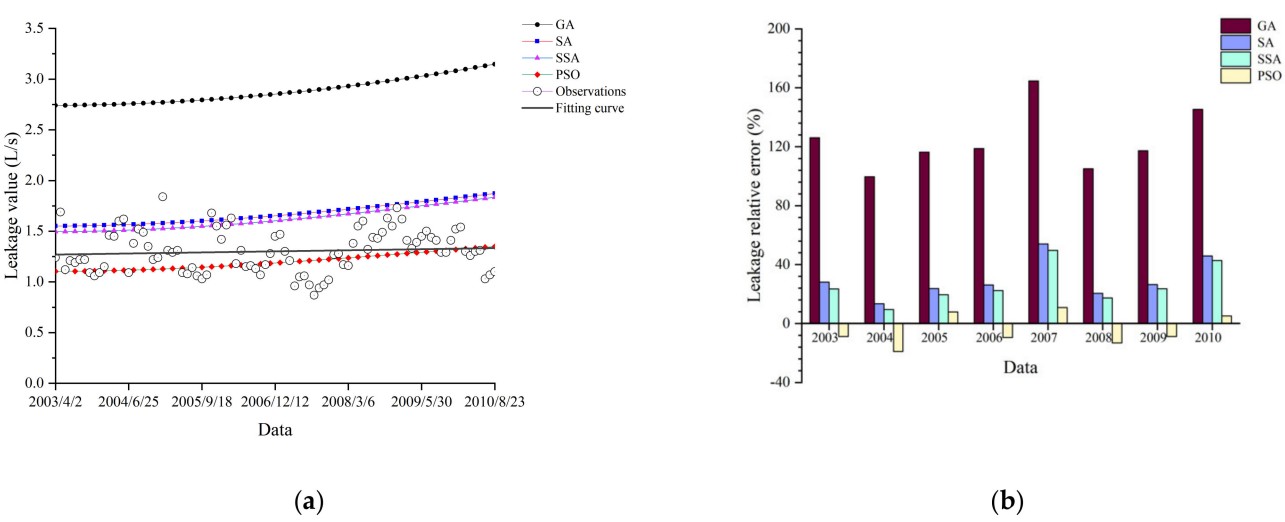

(**a**)  (**b**)

**Figure 6.** Comparison of measured and simulated values at the dam foundation: (**a**) leakage; (**b**) leakage relative error.

Relative error values were calculated using the annual average dam base leakage from the water measuring weir. Figure 6b shows the histogram of the relative error in dam base leakage determined by these algorithms from 2003 to 2010. The relative error of leakage simulated by GA is much larger than that of the other three groups. The maximum value of the relative error in dam foundation leakage is 170% in 2007. The results simulated by PSO demonstrate the most excellent performance. The leakage relative error of PSO is basically between −10% and 10%.

In the simulation of hydraulic head and leakage, the performance of GA is worse than that of PSO. This may be the result of the high dimensions of the model. It is challenging to meet the expected requirements in convergence accuracy by GA. The PSO strategy requires

no complex operations such as selection, crossover, and mutation, which adapts to this problem's solution better and achieves the global optimal solution more quickly.

### 5.3.3. Permeability Coefficient of the Impervious Curtain

Considering the excellent results of the PSO simulation in terms of head and leakage volume, the change in curtain permeability coefficient by PSO was analyzed to justify the simulation further. Figure 7a demonstrates the variation in curtain porosity for leaching times of 25, 50, and 100 years, respectively. The contour distribution of porosity shows that the calcium leaching effect leads to greater porosity on the upstream side of the curtain than on the downstream side and greater porosity in the upper part than in the middle and lower parts. The difference in porosity distribution intensifies with increasing leaching time. Figure 7b shows the porosity evolution at the height of 63 m for leaching times of 25, 50, and 100 years. After a century of leaching time, the porosity on the upstream side of the curtain is 0.206, with an increase of 0.056 compared to the porosity at the initial moment. The porosity on the downstream side is 0.170, increasing 0.020 in comparison with the porosity initially. Figure 7c presents the porosity distribution at different elevations of the curtain (at the elevation of 54 m, 63 m, and 72 m) after 50 years of leaching time. There is little difference in porosity distribution at the elevation of 54 m and 63 m. The upstream and downstream side porosity at the elevation of 72 m is 0.018 greater than the other two locations. The higher the altitude, the more significant the difference in the porosity distribution.

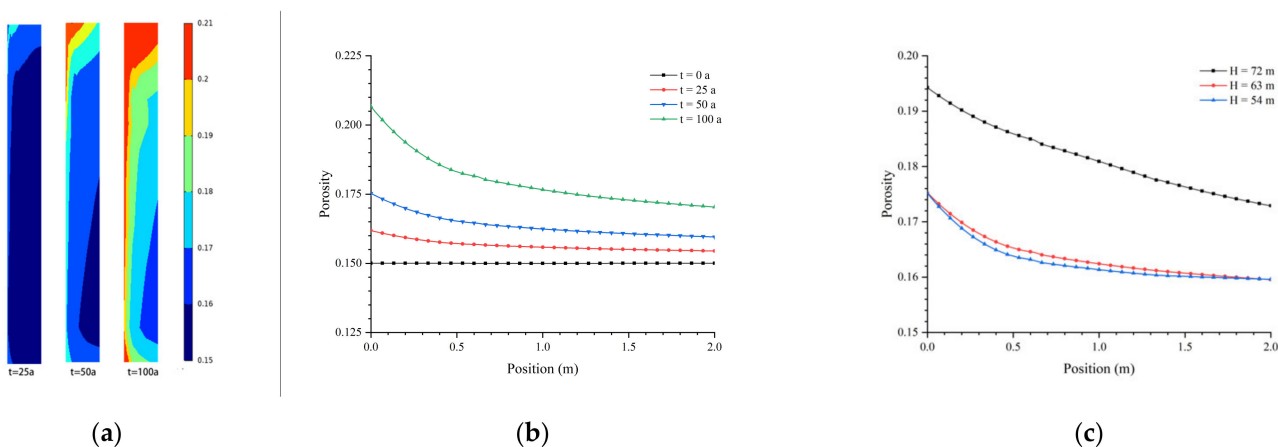

**Figure 7.** Impervious curtain porosity evolution in the leaching process: (**a**) porosity distribution; (**b**) porosity evolution at 63 m elevation in a century; (**c**) porosity evolution of three positions.

Figure 8a shows the distribution of the curtain permeability coefficient at 25, 50, and 100 years of leaching time, respectively. Similar to porosity, the permeability coefficient on the upper and upstream sides of the curtain increases significantly compared to other areas. The distribution difference of different parts is more evident with the increase of leaching time. Figure 8b shows the influence of different leaching times on the distribution of the curtain permeability coefficient at the altitude of 63 m. After 100 years of evolution, the permeability coefficient of the upstream side of the curtain is $2.60 \times 10^{-6}$, two orders of magnitude higher than the initial permeability coefficient. The permeability coefficient on the downstream side of the curtain is $2.72 \times 10^{-7}$, with an increase in one order of magnitude. Figure 8c presents the distribution of the curtain permeability coefficient at different elevations after the leaching time of 50 years. The permeability coefficient of the upstream side of the curtain at the elevation of 72 m is $1.45 \times 10^{-6}$, while that of the downstream side is $3.45 \times 10^{-7}$. The permeability coefficient at the altitude of 72 m is almost half an order of magnitude higher than the other two elevations. The higher the elevation, the more pronounced the increase in the permeability coefficient of the impervious curtain.

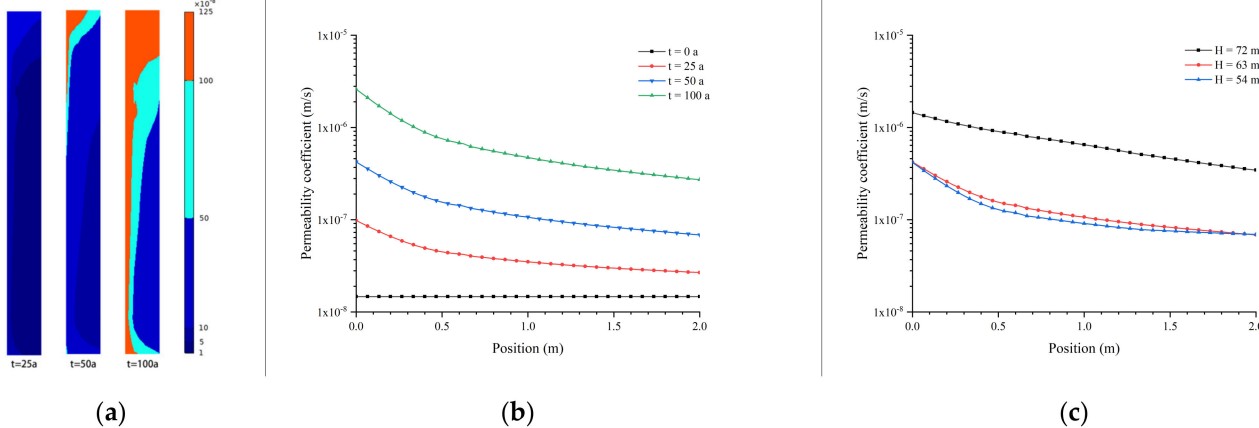

**Figure 8.** Impervious curtain permeability coefficient evolution in the leaching process; (**a**) permeability coefficient distribution; (**b**) permeability coefficient evolution at 63 m elevation in a century; (**c**) permeability coefficient evolution of three positions.

## 6. Discussion

The permeability coefficient of the curtain increases with the operating time, affecting the benefit and safety of water conservancy projects. In this paper, the inverse analysis method is introduced into solving the parameters of curtain calcium leaching, which solves the problem that curtain samples are not easy to obtain and leaching parameters are not easy to determine. This method can be used to predict the permeability coefficient of the curtain for many years.

Based on ELM and four optimization algorithms, accurate calculation results are obtained. The calculated leaching parameters are used for finite element forward analysis. The results show that PSO has the best simulation performance. This may be due to the multi-dimensional nature of the model. Compared with other optimization algorithms, the PSO strategy is simple to operate and can better adapt to the solution of this model so as to obtain the global optimal solution more efficiently. It is suggested that ELM and PSO be combined to obtain reasonable simulation results.

The simulation results of the permeability coefficient show that the permeability coefficient of the upstream side and upper part of the curtain increases more significantly than that of other areas. These areas need to be intensely focused on and monitored to prevent leakage caused by the calcium leaching effect, which is of great significance for the safety of the project.

## 7. Conclusions

Aiming at the calcium leaching model in porous media, this study proposed the inverse method into the calcium leaching parameter solution process. Then, a back analysis for permeability coefficients of the curtain at Shimantan gravity dam was realized. The objective function was constructed from time-series measurements of hydraulic head and leakage. Finite element analysis was performed to construct neural network learning samples. ELM was applied to build a non-linear mapping of leaching parameters and error values. Four optimization algorithms were devoted to accelerating the computation process. The hydraulic head and leakage errors of the positive analysis for the four optimization algorithms were compared, and the trend of the curtain's permeability coefficient in 100 years was analyzed. The reliability and uniqueness of the inverse results were improved. The main conclusions of the paper are indicated as follows:

(1)    Based on ELM and four optimization algorithms, the inversion values of leaching parameters by four algorithms show small differences. The results are all within the parameter range.

(2) Four sets of leaching parameter results were used in the positive analysis. Among the four results, the curves fitted by PSO corresponding to the leaching parameters are in the best agreement with the measured values and show the highest prediction accuracy, which indicates that the inversion method is reliable and effective.

(3) The values of permeability coefficients on the upper and upstream sides are greater than in other areas, showing that these areas are the most vulnerable parts of the grout curtain. Focusing on these vulnerable parts and strengthening safety management is necessary for the safety of water conservancy projects. The results illustrate the feasibility of the inversion analysis for obtaining the calcium leaching parameters under advection-diffusion-driven leaching processes.

**Author Contributions:** Conceptualization, Y.S. and L.X.; methodology, Z.S.; software, Y.S.; validation, K.Z. and C.Y.; formal analysis, L.X.; investigation, K.Z.; resources, Z.S.; data curation, C.Y.; writing—original draft preparation, Y.S.; writing—review and editing, Y.S.; visualization, K.Z.; supervision, Z.S.; project administration, Z.S.; funding acquisition, L.X. All authors have read and agreed to the published version of the manuscript.

**Funding:** This research was funded by "National Natural Science Foundation of China, grant number 52179130", "Postgraduate Research & Practice Innovation Program of Jiangsu Province, grant number KYCX18\0598", and "Fundamental Research Funds for the Central Universities, grant number 2018B630\X14".

**Conflicts of Interest:** The authors declare no conflict of interest.

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
