# Peer review of "Inversion Analysis of Impervious Curtain Permeability Coefficient Using Calcium Leaching Model, Extreme Learning Machine, and Optimization Algorithms"

_applsci, doi:10.3390/app12073272_

Round 1

Reviewer 1 Report

The inverse analysis strategy was used to study the parameters of the calcium leaching effect.  Combining FEM with time series data, Genetic algorithm (GA), simulated annealing (SA), sparrow search algorithm (SSA), and particle swarm optimization (PSO) were used to accelerate the iterative search for the target parameters.  The results show good agreement with PSO.   This research is a good example of applications of  optimization method.    Presentations and discussion are well presented.  

A few errors and be easily corrected.

1)  systems of equations (1), (2), ...  comma are missing 

2)  more on methods like GA SA SSA PSO.  What are parameters used in the experiments?  population sizes.. particle size.. mutation rate?  velocity equations? etc.. 

3) Hope to see a discussion why PSO is better than the others in this particle problems structure. 

Reviewer 2 Report

The article presents an approach of inversion analysis of impervious curtain permeability coefficient using a combination of calcium leaching model, ELM, and optimization. The strategy used in the study is feasible. The article is written in an acceptable manner and has valuable content for the Journal's audience. The article is simple but concise and thorough and should be considered a good read as well. Research design is not questionable and methods are well described. It appears that simulation, optimization, and other calculations have been performed correctly.

Please correct:

205-326 lines seem that they are not presented in a "justified" text style, please adapt and unify style according to Guidelines

274 - "0.1m" - spacing missing

296 - Table 2. -> Value column "25.40kN/m3" and "23.51kN/m3" spacing missing

304 - Table 4. -> superscript is missing

346 - "algorithms from 2013 to 2010." -> 2003 to 2010?

358 - uppercase the first letter
